# Simulating emergence of novelties using agent-based models

**Mikihiro Suda** , **Takumi Saito, Nanami Iwahashi, Ciaran Regan** , **Mizuki Oka** *

Grad. School of Science and Technology, University of Tsukuba, Tsukuba, Ibaraki, Japan

* mizuki@cs.tsukuba.ac.jp

**Data Availability Statement:** All the datasets supporting the findings of this work can be freely accessed through online databases, indicated within the manuscript. Moreover, a complete collection of the datasets can be downloaded from

## Abstract

Understanding the growth and evolution of social networks is an important area of study, as these networks form the foundation for many popular online services such as social networking sites (SNS) and online games. However, previous models developed to explain the growth mechanisms of these networks have struggled to accurately reproduce certain behaviors that are frequently observed in real data, such as waves of novelty, in which new individuals or topics receive more attention than existing ones for a short period of time. In this study, we introduce a new model that incorporates context information into existing agent-based models in order to more accurately capture the structure and growth dynamics of these networks. Context information is introduced through labels based on the timing of appearance and relationships with antecedent agents. New agents are first added to the network when they are called by existing agents, and at this time they are also given a label. Agents added to the network at the same time by the same agent will have the same label. These labels are used to classify agents and give them different selection probabilities. This newly introduced selection probability creates a mechanism in which new agents receive attention beyond preferential attachment. By comparing the results of our model with real data on ten metrics, we demonstrate that it is able to produce behavior that more closely resembles real data. This improved understanding of the dynamics of social networks has important implications for designing effective interventions, including strategies for user acquisition and retention.

## Introduction

Human society is characterized by a diverse range of interactions, including casual greetings and conversations with friends, organizational transactions and negotiations, and even treaties between nations. The widespread adoption of the internet and smartphones has facilitated the documentation and accumulation of data on these interactions, enabling researchers to model human interactions and identify their underlying commonalities and principles.

Mathematical models have been widely employed as a means of replicating and understanding human activities. In particular, it has been demonstrated that hashtags, a prevalent feature of social media networks, can be accurately modeled using stochastic process models

the public repository https://figshare.com/articles/
dataset/Emergence_and_evolution_of_social_
networks_through_exploration_of_the_Adjacent_
Possible/13308428. All the data in this work has
been collected according to the terms of services
of the corresponding websites and can be freely
shared. None of the datasets about the online
activities has been initially collected by the authors
and has been accessed from other public
repositories.

**Funding:** This work was supported by JSPS
KAKENHI Grant Numbers 19H04214, 20H04163,
21H03414. There was no additional external
funding received for this study. The funders had no
role in study design, data collection and analysis,
decision to publish, or preparation of the
manuscript.

**Competing interests:** The authors have declared
that no competing interests exist.

such as Polya's urn model [1–4] and the Yule-Simon model [5–11], which feature stochastic novelty and preferential attachment.

While these models are able to reproduce macro statistical laws at the population level, they have been shown to be insufficient at reproducing the local microstructural interactions of human activity [12]. The incorporation of random novelties and preferential attachment alone is not sufficient to capture the open-ended nature [13–15] of human activity such as *waves of novelty*, the emergence of new ideas, technologies, or trends that spread rapidly through a population or system, often replacing or supplanting existing ones [16–19].

There have been a number of models proposed to more accurately capture the phenomena of novelties in real data. One such model is based on the concept that novelty does not emerge randomly, as is often assumed in existing generative models, but rather from the *adjacent possible space*. The concept of the adjacent possible space refers to the realm of possibilities that are a step away from what currently exists, but which may soon become a reality. This idea was initially theorized by Stuart Kauffman in order to explain the evolution of molecules and organisms [20]. A similar concept can be found in the protein space theory proposed in [21], which argues that gene evolution occurs through the accumulation of minute changes to existing genes, subject to the limitation that these changes must result in viable phenotypes. Kauffman expanded and generalized this theory to apply not only to the evolution of genes, but also to the evolution of human relationships and other areas.

Kauffman's claim, that novelty emerges from the adjacent possible space, was mathematically formulated and modeled by Tria et al. [4], with the introduction of the Urn Model with Triggering (UMT). In the UMT, colored balls are drawn sequentially from a single urn at random. When a new ball is extracted for the first time, a number of new balls are placed into the urn, expanding the space of possibilities for the next iteration. The sequence of extracted balls represents the interaction history of the system. With a focus on modelling social systems, Ubaldi et al. extended this model to an agent-based system, allowing each agent to have their own unique urn [22]. In this context, the agent represents an individual in the social system, with their urn representing the space of possible interaction partners for that individual. Every ball in the urn has an ID representing another individual, and the selection of an urn and a ball from inside that urn constitutes an interaction between the individual represented by the urn and the individual represented by the ID of the ball. This model has successfully reproduced several real-world human behaviors with improved accuracy. For instance, the model can replicate the dynamics and structure of various social networks, including Twitter's network of username mentions, co-authorship networks for papers, and outgoing cellular networks.

However, the models proposed by Tria et al. and Ubaldi et al. fail to reproduce the *waves of novelty* typically observed in real data [12]. The frequency at which novelties appear in these models significantly differs from that seen in real data. Additionally, both models exhibit a bias in which agents that appear earlier in the process continue to attract more attention than agents observed in real data. This preferential attachment is stronger in these models compared to real data, as the probability of selecting an agent is proportional to the number of times it appears. Previous studies have demonstrated that adding an adjustable bias between retracing the past and looking towards the future in agent selection results in a model that more closely matches real behaviour. In particular, Monechi et al. generalized UMT by introducing classes, labels and a collective effect on how the space of possibilities is explored, which modifies the probability of agent selection [19]. These changes were shown to accurately reproduce the patterns of popularity observed in empirical data. In an attempt to extend this approach to an agent-based model, Suda et al. introduced a Generalised Agent Based Model (GABM), extending the Ubaldi et al. model. In this GABM, agents are assigned a class based on the number of balls in their urn. At caller selection time, a class is selected with a probability

proportional to the number of urns in that class and subsequently, an agent is selected from the class based on a predefined strategy. Although this GABM allowed new agents to receive increased attention, it failed to replicate waves of novelty, with older agents always being the most active agents in the network.

This paper presents a model that improves upon the ability to capture the behavior of real social systems by introducing labels and biases in the selection of agents in the agent-based model proposed by Ubaldi et al. Contextual information is incorporated through labels based on the timing of appearance and relationships with antecedent agents. New agents are initially added to the network when they are called by existing agents, and at this time they are also assigned a label. Agents added to the network at the same time by the same agent will have the same label. These labels are used to classify agents and give them different selection probabilities, introducing a mechanism by which new agents receive attention beyond preferential attachment. The proposed model outperforms both the model proposed by Ubaldi et al. and the model proposed by Suda et al. in accurately reproducing real data behavior, specifically through the successful generation of waves of novelty. The enhanced understanding of the dynamics of social networks has important implications for the design of effective interventions, including strategies for user acquisition and retention.

## Methods

Individuals partake in a multitude of social interactions, including verbal communication, mentions on social networking sites, and co-authorship of articles. These interactions can be characterized by the distinction between the initiator, referred to as the caller, and the recipient, referred to as the callee. For instance, the individual who initiates a telephone call serves as the caller, while the individual who receives the call serves as the callee. The proposed model endeavors to replicate human interaction by selecting successive pairs of caller and callee agents from the population under examination.

### Agent-based Polya's urn model by Ubaldi et al

In Polya's urn model, preferential attachment is achieved through a process of randomly selecting a ball from an urn containing balls of various colors, incrementing the number of balls of that particular color by a factor of $\rho$, and subsequently returning the balls to the urn [1]. While many existing models based on Polya's urn model posit that all balls share a single urn [4, 19], this assumption can be problematic, particularly when modelling social systems where each individual or agent represented by the balls have access to a separate space of information. To address this limitation, Ubaldi et al. proposed an agent-based model in which each agent in the social system has a distinct urn with its own information space [22]. This extension of Polya's urn model was shown to improve the accuracy of reproducing interactions observed in real data.

In the Ubaldi et al. model, each agent is assigned a unique identifier and possesses an individual urn. These urns contain balls with IDs that represent other agents with whom the agent has interacted in the past and potential agents with whom the agent may interact in the future. The number of balls in each urn is variable and reflective of the characteristics of the agent. The model outlined by Ubaldi et al. comprises the following three steps for agents to interact and form a network:

Step 1. Select a caller (interaction initiator) agent from the population.

Step 2. Select a callee (interaction partner) agent from the caller agent's urn.

Step 3. Facilitate an interaction between the caller and callee agents, resulting in the formation of an edge linking them.

These three steps constitute one iteration. This process is repeated to generate a time series of events and a network of interactions between agents. The present study extends the agent-based model proposed by Ubaldi et al. by providing further details of the model, its limitations, and the proposed extensions.

**Step 1: Select a caller agent from the population**

In step one, a caller agent is randomly selected from the population with probability proportional to the urn size of the agent. The urn size serves as a weighting factor, such that an agent with an urn containing ten balls is ten times more likely to be selected than an agent with an urn containing one ball.

**Step 2: Select a callee agent from the caller agent's urn**

In step two, a callee agent is randomly selected from the caller agent's urn.

**Step 3: The interaction between the caller and callee agent**

Step three involves the interaction between the caller and callee agent. The interaction is defined by three parameters: $\rho$, $v$, and $s$. First, the caller and callee agent add $\rho$ balls bearing the ID of the interaction partner into their urn. That is, the caller gains $\rho$ copies of the callee agents ID into its urn and the callee gains $\rho$ copies of the caller agents ID into its urn. This operation can be represented as an increase in the weight of the edge between the caller and callee agents by $\rho$. Next, if this is the first time that the caller and callee interact, both agents select $v + 1$ balls from their partner's urn, based on a predefined strategy $s$, and add them to their own urns. This operation can be represented as the expansion of the adjacent possible space.

The parameter $\rho$ adjusts the size of preferential attachment, with larger values leading to a higher probability of interacting with the same agent. The parameter $v$ adjusts the magnitude of novelty acquisition, with larger values leading to a greater expansion of the adjacent possible space.

The parameter $s$ specifies the search strategy for new agents in the adjacent possible space. There are numerous possible strategies for $s$, such as randomly selecting agents or selecting the most frequently selected agents. Previous studies have shown that varying $s$, as well as $\rho$ and $v$, can reproduce behavior similar to real data [12, 22, 23].

Finally, the addition of new agents to the population occurs when an agent is selected as a callee for the first time. The event of being selected as a callee agent for the first time corresponds to a situation where a person is mentioned on a social network for the first time, a person becomes a co-author of a paper for the first time or a person is called on a mobile phone for the first time, etc. In Ubaldi et al.'s model, an agent that becomes a callee for the first time creates $v + 1$ new agents with empty urns and adds those agents IDs to its own urn.

## Generalised agent-based Polya's urn model by Suda et al

The version of Polya's urn model proposed by Ubaldi et al. has demonstrated its ability to capture network structure and growth dynamics as well as Zipf's and Heaps' laws. However, as outlined in the Results section, this model fails to reproduce the phenomena of *waves of novelty* observed in real-world data. This limitation arises due to the selection probability of caller agents being based solely on urn size, resulting in preferential attachment having a stronger influence than what is observed in real-world data, and agents that appear early on continue to attract more attention. In an effort to address this problem, Suda et al. introduced a

Generalised Agent Based Model (GABM) [12]. This GABM alters the caller agent selection in step 1 of the Ubaldi et al. model, in an attempt to reduce the strength of preferential attachment and allow for waves of novelty.

In this approach, each agent $i$ is assigned to a class $c_n$ based on it's previous occurrence frequency $n_i$. For example, an agent who has interacted twice would belong to $c_2$. First, a class $c_n$ is selected randomly in proportion to the total occurrence of agents in each class. That is, if ten agents belong to $c_1$ and two agents belongs to $c_2$ then $c_1$ is five times more likely to be selected. Subsequently, an agent is selected from $c_n$. This selection of an agent from a class is dependent on a predefined strategy $s_c$. Suda et al. suggest a last-in-first-out strategy and a piggyback strategy with the goal of simulating waves of novelty. This GABM can then be defined by the three parameters from the Ubaldi et al. model and an additional caller selection strategy parameter $s_c$.

## Proposed model

Although the GABM introduced by Suda at al. allowed for newer agents to receive an increase in attention when compared with the Ubaldi et al. model, it still failed to reproduce waves of novelty, with early occurring agents being the most popular throughout the network history. To overcome this limitation, a new mechanism is needed to change the selection attachment of caller agents and introduce *waves of novelty* in which new agents attract attention beyond preferential attachment. Inspiration for addressing this issue can be found in the work of Monechi et al., where the introduction of "labels" is proposed as a means to introduce contextual information, such as timing of appearance and relationships with antecedent agents, into the model.

The proposed model in this study extends the model of Ubaldi et al. by introducing a label for each agent. In this model, a label is assigned to newly-born agents, which does not change once assigned. That is, when an agent in the system is selected as a callee for the first time, the $v + 1$ new agents that are added to the system will all be assigned to the same label. There is a one-to-many relationship between labels and agents, meaning several agents can have the same label. In particular, there will always be $v + 1$ agents who all share the same label. Unlike the model of Ubaldi et al., where the selection probability of the caller agent is determined by the size of the agent's urn, in the proposed model, the selection probability is determined by the agent's label.

In the proposed model, agents are classified into classes based on their labels and selection history. In particular, we split the agents into five classes to distinguish between agents who have (not) been selected as callers before and agents who have (not) been selected as callers recently. By creating this distinction, we can control the selection probability of recently selected callers with the goal of simulating waves of novelty. Each class is assigned a different weight, resulting in variation in the caller selection probability of the agent. The five classes are as follows:

Class 1: An agent with an empty urn.
  : weight 0

Class 2: An agent with the label $L_{t-1}$ and has been selected as a caller at least once.
  : weight 1

Class 3: An agent with the label other than $L_{t-1}$ and has been selected as a caller at least once.
  : weight $\zeta f(N_{L_{t-1}}, N_{\overline{L_{t-1}}})$

Class 4: An agent with label $L_{t-1}$ and has never been selected as caller.
  : weight $g(N_{L_{t-1}}, N_{\overline{L_{t-1}}})$

Class 5: An agent with label other than $L_{t-1}$ and has never been selected as a caller.
: weight $\eta g(N_{L_{t-1}}, N_{\overline{L_{t-1}}})$,

where $t$ indicates the current iteration, $L_{t-1}$ indicates the label which is selected at iteration $t-1$, $N_{L_{t-1}}$ indicates the number of agents who have label $L_{t-1}$, $N_{\overline{L_{t-1}}}$ indicates the number of agents who do not have label $L_{t-1}$. It is worth highlighting that $N_L$ will always be $v + 1$, as the number of new agents created with the same label is always $v + 1$ and $N_{\overline{L}}$ is always equal to the number agents who have previously interacted, less the $v + 1$ agents with label $L$. We define functions $f, g$ as follows:

$$f(N_L, N_{\overline{L}}) = \frac{N_L}{N_L + \zeta N_{\overline{L}}} \qquad (1)$$

$$g(N_L, N_{\overline{L}}) = \frac{N_L + \zeta f(N_L, N_{\overline{L}}) N_{\overline{L}}}{N_L + \zeta N_{\overline{L}}}, \qquad (2)$$

where the $\zeta$ and $\eta$ are parameters of the model, both of which take values between 0 and 1.

This choice of $f, g, \zeta$ and $\eta$ are motivated by the work of Monechi et al. [19], who have shown that with this choice, the growth of the number of distinct elements with time $t$ follows Heap's law. The parameters $\zeta$ and $\eta$ modulate the propensity to exploit the past and the propensity to explore the new, respectively. This can be seen if we consider the limits of the parameters, which are shown in Table 1.

With $\zeta = 0, \eta = 0$ there is an equal probability of selecting an agent in class 2 or class 4. That is, there is an equal likelihood of selecting an agent who has or has not been selected as a caller before. As $\zeta$ is increased to $\zeta = 1, \eta = 0$, the selection probability of class 3 increases, increasing the likelihood of selecting an agent who has already been selected before and hence increasing the propensity of the system to exploit that past. In the case where $\zeta = 0$ and $\eta = 1$, class 4 and 5 both have weights 1, with the next caller likely to have never been selected before and the system therefore tending to explore the new. As $\eta$ is increased to $\zeta = 1, \eta = 1$, the weight of class 4 and 5 decreases and the weight of class 3 increases. At large $N_{\overline{L}}$, class 4 and class 5 are twice as likely to be selected as class 3, but the system is most likely to choose class 2, which is recently selected past callers.

The difference between this proposed model and both the original agent-based model proposed by Ubaldi et al. and the generalised agent-based model proposed by Suda et al. is that the caller selection rate in Step 1 is based on label weights. Step 2 and Step 3 of the proposed model is identical to that of the other two models.

A conceptual diagram of the proposed model can be found in Fig 1, where the results of three iterations after the initial state are shown. In the first iteration, the caller is chosen randomly as no events have occurred yet, but from the second iteration onwards, the caller is chosen by label-based weighting.

**Table 1. The weights of the five classes for given parameters.**

| $\zeta$ | $\eta$ | $w_1$ | $w_2$ | $w_3$ | $w_4$ | $w_5$ |
|---|---|---|---|---|---|---|
| 0 | 0 | 0 | 1 | 0 | 1 | 0 |
| 1 | 0 | 0 | 1 | $\frac{N_L}{N_L + N_{\overline{L}}}$ | $\frac{N_L + \frac{N_L}{N_L + N_{\overline{L}}} N_{\overline{L}}}{N_L + N_{\overline{L}}}$ | 0 |
| 0 | 1 | 0 | 1 | 0 | 1 | 1 |
| 1 | 1 | 0 | 1 | $\frac{N_L}{N_L + N_{\overline{L}}}$ | $\frac{N_L + \frac{N_L}{N_L + N_{\overline{L}}} N_{\overline{L}}}{N_L + N_{\overline{L}}}$ | $\frac{N_L + \frac{N_L}{N_L + N_{\overline{L}}} N_{\overline{L}}}{N_L + N_{\overline{L}}}$ |

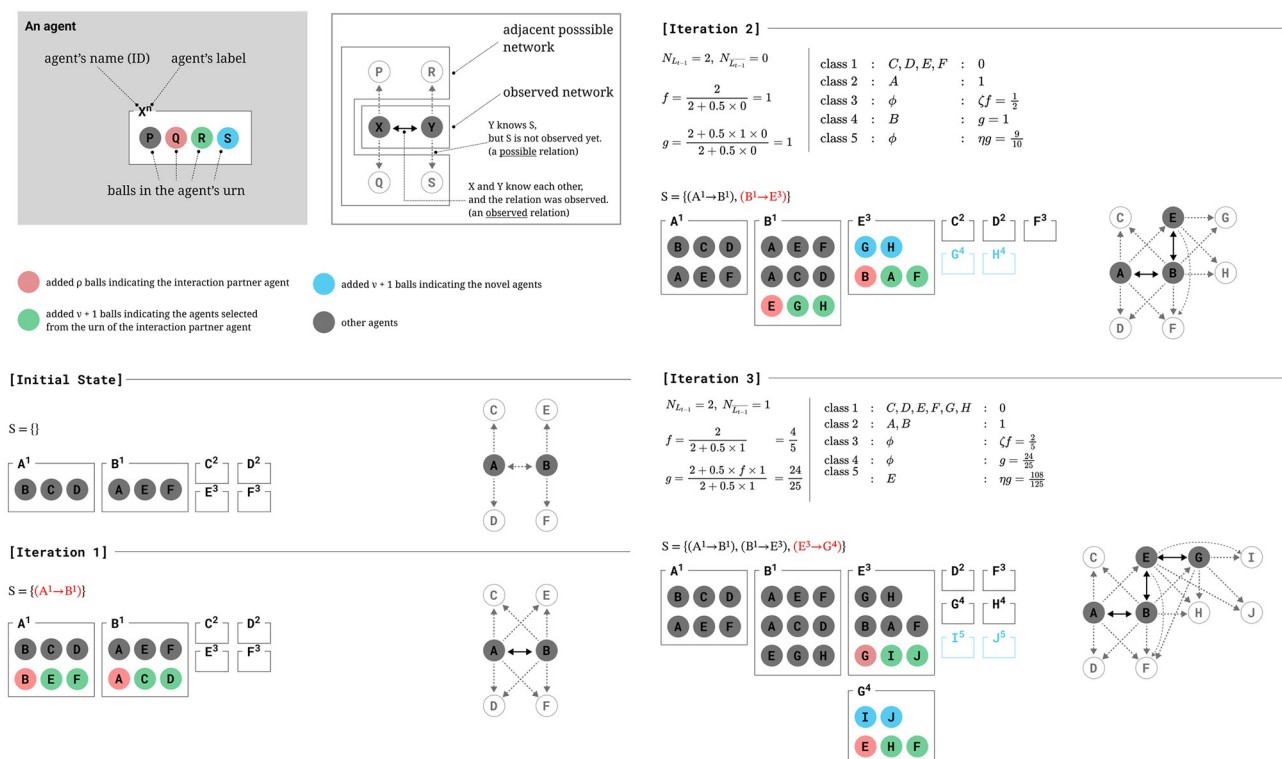

**Fig 1. An example flow of the proposed model. General:** Each agent in the system is characterized by three attributes: a unique name or identifier (ID), a label, and an urn. The urn contains balls bearing the IDs of other agents. The diagram illustrates the system, with red balls representing the $\rho$ agents that need to be added to the interaction partner's urn, green balls representing $\nu + 1$ agents selected from the interaction partner's urn based on strategy $s$, and blue balls indicating new agents that will be added when they are selected as caller agents for the first time. The network comprising these agents is comprised of both edges between agents that have already interacted, represented by solid lines, and edges between agents that have not yet interacted but are present in the urn, represented by dotted lines. **Initial State:** This example considers the initial state of the system with the following parameter values: $\rho = 1$, $\nu = 1$, $\varsigma = 0.5$, and $\eta = 0.5$. The initial state of the system comprises of two agents, $A^1$ and $B^1$, each possessing an urn. Additionally, there are four empty urns, designated as $C^2$ and $D^2$, and $E^3$ and $F^3$. The urn of Agent $A^1$ contains the ball bearing the ID of its interaction partner, Agent $B^1$, as well as $\nu + 1$ agents without urns (in this case, $C^2$ and $D^2$). Similarly, the urn of Agent $B^1$ contains the ID of its interaction partner, Agent $A^1$, and $\nu + 1$ additional agents without urns (in this case $E^3$ and $F^3$). **Iteration 1:** The first iteration of the process begins by randomly selecting a caller agent. In this example, Agent $A^1$ is selected as the caller agent. The callee agent is then chosen by selecting one agent (represented by a ball with an ID) from the caller agent's urn, in this case, Agent $B^1$ is selected as the callee agent. Following this, the caller agent's urn is filled with $\rho$ callee agents and $\nu = 1$ additional agents that are chosen randomly from the callee agent's urn (represented by green balls in the diagram). Similarly, the callee agent's urn is filled with $\rho$ caller agents and an additional $\nu = 1$ agents (also represented by green balls in the diagram) chosen randomly from the caller agent's urn. This interaction is then represented as a solid edge between Agent $A^1$ and Agent $B^1$ in the network diagram, forming an observed network, while the other agents form adjacent possible networks. **Iteration 2:** The second iteration of the process employs a mechanism for selecting the caller agents based on label weights assigned to each agent. As observed in the previous iteration, there are two agents, $A^1$ and $B^1$, possessing the label 1 of the previous caller agent $A^1$, resulting in $N_{L_{t-1}} = 2$. Since there are no agents with labels other than 1, $N_{\overline{L_{t-1}}} = 0$. Using these values and the expressions presented in Eqs 1 and 2, the values of f and g as shown in the diagram are obtained. The agents in the population are then classified into classes, and the weights of $C^2, D^2, E^3, F^3$ are calculated as zero, while the weights of $A^1, B^1$ are calculated as 1. Based on these weights, the caller agent $B^1$ is selected, and the callee agent $E^3$ is selected from its urn. Since $E^3$ is selected as a callee agent for the first time, $\nu = 1$ new agents are added to its urn, as represented by the blue balls in the diagram. The subsequent operations are identical to those in the first iteration, as represented by the red and green balls in the diagram. **Iteration 3:** In the following iteration, the same procedure for selecting caller and callee agents based on their label weights, as previously outlined in Iteration 2, is applied. In this iteration, the agent $E^3$ is selected as the caller agent according to the calculated weight, and the callee agent is chosen as $G^4$ from $E^3$'s urn.

## Evaluation

The evaluation of the models presented in this study is based on six metrics proposed by Ubaldi et al. These metrics are designed to assess the topology and dynamics of the network generated by the model as described in the study of Ubaldi et al. [22]. Furthermore, the evaluation also employs four novelty metrics proposed by Monechi et al. [19]. It should be noted that

while Ubaldi et al. proposed eight metrics, two of them were not provided with sufficient detail and were not reproducible, thus they are not included in the metrics used in this paper.

**Metrics regarding network dynamics and topology.** *Exponent of Heaps' law.* Previous research has established that activity logs for a variety of social network services and human activities exhibit a pattern consistent with Heaps' Law [24]. Therefore, evaluating the extent to which the data generated by the model under study adheres to this pattern, as observed in real-world data, is an important indicator of its performance.

To this end, the degree of agreement of the exponent $\gamma$ of Heaps' Law is employed as a metric. Specifically, this is accomplished by comparing the relationship between the number of edges $A(t)$ and the number of interaction events $t$ as represented by the following equation:

$$A(t) \propto t^{\gamma}, \tag{3}$$

where $\gamma$ is the exponent characterizing the growth of the number of edges. Utilizing this metric allows for the assessment of how well the model generates network dynamics that resemble those observed in real-world social networks in terms of the growth of the number of edges.

*Clustering coefficient.* The clustering coefficient is a measure of the clustered nature of the network. The clustering coefficient $C_i$ of a node $i$ in the network is defined as follows:

$$C_i = \frac{2e_i}{k_i(k_i - 1)}, \tag{4}$$

where $k_i$ and $e_i$ denote the number of neighboring nodes and edges of node $i$, respectively. The clustering coefficient $C$ for the entire network is obtained as the average of the individual node clustering coefficients $C_i$. Barabasi et al. have established that the clustering coefficient $C$ of many social networks falls in the range of $0.1 \leq C \leq 0.7$ [25].

*Novelty and closure of interactions.* When comparing models with real-world data, a key aspect to consider is the topology of the final generated and growing network. In line with the approach proposed by Ubaldi et al. [22], the novelty and closure of the interactions (i.e., edges) are used as metrics to evaluate the topology.

The novelty of an interaction refers to whether it has occurred previously in the network or not, with interactions that have occurred previously being denoted as `old`, and those that have not as `new`. The closure nature of an interaction, on the other hand, refers to whether the interaction took place between two nodes that have common neighbors; interactions that take place between nodes with common neighbors are `closed`, whereas those that do not are `open`.

These two characteristics are combined to classify all interactions into four types: `old-closed`, `old-open`, `new-closed`, and `new-open`. The occurrence frequency of these four interaction types normalised by the total number of interactions, represented by *OC*, *OO*, *NC*, and *NO* respectively, is used as a metric to evaluate the topology of the network growth process. This enables an assessment of how well the model is able to generate network structures that resemble those observed in real-world data in terms of the novelty and closure of interactions.

**Metrics for novelty production.** To evaluate the production of novelty, as an aspect of the model's performance, this study employs the four indicators proposed by Monechi et al. [19]. These indicators are the youth coefficient, recentness, local entropy, and Gini-like coefficient, which provide insight into the distribution and patterns of new interactions and agents in the generated network over time. Together, they provide a comprehensive understanding of the model's ability to generate novelty dynamics and structure in the network.

*Youth coefficient.* The growth of an open-ended network necessitates the attraction of appropriate attention to nodes that are in their nascent stages of development, as well as to those that have emerged in later stages of growth. To measure this property, the youth coefficient, denoted as $Y$, is employed as a metric.

In order to calculate the youth coefficient, the growth time step of the network, represented by $T$, is divided by a constant width, denoted as $\Delta\tau$. Let $B(I_i)$ (where $1 \leq i \leq \frac{T}{\Delta\tau}$) be the average birth step of the nodes that interacted during interval $I_i$. The relationship between $I_i$ and $B(I_i)$ can be plotted and if the slope of this plot is denoted as $\lambda$, the youth coefficient $Y$ is defined as follows:

$$Y = \frac{\lambda}{\Delta\tau}. \tag{5}$$

It is worth noting that a higher value of $Y$ indicates that new nodes in the network are attracting more attention during each interval, that is to say, they are receiving a higher level of engagement within the system.

*Recentness.* The metric of recentness, denoted by $R$, serves to indicate if active nodes first appeared early or late in the network history. As $R$ approaches 0, it implies that the most active nodes appeared early in the network history, conversely, if $R$ approaches 1, it implies that the most active nodes appeared late in the network history.

Similarly to the youth coefficient, the recentness is calculated by considering the interval $I_i$ and $t_{max}(I_i)$, which represents the first appearance time of the node with the highest number of interactions in $I_i$. The recentness $R$ is mathematically defined as follows:

$$R = \frac{\sum t_{max}(I_i)}{\sum i\Delta\tau}. \tag{6}$$

*Local entropy.* The local entropy serves as a measure of the diversity of the agents that interact within a given interval $I$. High local entropy, denoted as $h$, is observed when the network growth exhibits a mixture of agents that are born early within each interval, as well as agents that appear later in the network's development.

To calculate the local entropy, let $n_j$ represent the number of interactions of a particular node $j$ that occur within interval $I_i$. Then, if we define $f_j = n_j/\Delta\tau$, the local entropy of the interval, $h_i$ is mathematically defined as follows:

$$h_i = \frac{-\sum f_j \log f_j}{\log D(I_i)}, \tag{7}$$

where $D(I_i)$ is the number of distinct nodes that interacted within the interval $I_i$. The local entropy of the network growth process, denoted as $\langle h \rangle$, is obtained as the average of the local entropy of each interval.

It is worth noting that entropy measures the degree of disorder within a system. As such, the local entropy is indicative of the degree of disorder within the network regarding the diversity of the agents that interact in each interval. The average of all local entropy in the network growth process, gives a broader view of how the diversity of the agents evolves over time.

*Gini-like coefficient.* The Gini coefficient, a widely utilized metric in economics, is used in this study as a measure of the disparity in the degrees of attention received by individual nodes within the network.

To calculate the Gini-like coefficient, we first arrange the nodes in birth order and let $r_i$ represent the birth order of node $i$. We then define $x_i = r_i/D$, where $D$ is the total number of distinct nodes. Additionally, we define $a_i$ as the interaction frequency of node $i$, and let

$y_i = \sum_{k=1}^{i} a_k$. Plotting the relationship between $x_i$ and $y_i$ produces a Lorenz curve. The area bounded by this curve and the line $y = x$ is the Gini-like coefficient, denoted as $G$. As the disparity increases, the Lorenz curve moves further away from the straight line $y = x$, hence, a larger Gini coefficient $G$ indicates a greater disparity in the attention received by the nodes.

In order to evaluate the degree of adaptation of the proposed model to real-world data, we utilize the distance metric, $d$, which is defined as the sum of the absolute differences between the model-generated values and the corresponding empirical values for each of the 10 metrics described above. Mathematically, it is represented by the following equation:

$$d = \sum^{p \in P} |p_{model} - p_{empirical}|, \tag{8}$$

where $P = \gamma, C, OC, OO, NC, NO, Y, R, \langle h \rangle, G$ represents the set of metrics used in the evaluation, with each metric taking values in the range [0, 1].

To determine the best-fit values for the model's parameters $\rho$, $v$, $s$, $\zeta$ and $\eta$, a search through all possible combinations of parameter values within a specified range is conducted. By minimizing the distance metric $d$, the parameter values that best-fit the real-world data can be identified.

## Empirical data

In this study, we used two publicly available networks, the APS Co-authors Network and Twitter Mentions Network (TMN), as our evaluation data [26]. The APS Co-authors Network, which is obtained from the journal American Physical Society, consists of co-authorship relationships, where up to 10 people can simultaneously be associated. Following the approach of Ubaldi et al., we randomly extracted a pair of co-authorship relationships from a single article, and randomly selected one as the caller agent and the other as the callee agent.

The Twitter Mentions Network, which is a dataset of tweets from Twitter, was also used as our evaluation data. In this network, a user's mention of another user in their tweet is considered as an act of selecting a callee agent by a caller agent. To maintain its authenticity, we utilized the raw data of the Twitter Mentions Network, without any pre-processing.

## Code availability

The code for the proposed model and the experiment has been made publicly available on GitHub: https://github.com/tsukuba-websci/UrnBasedDynamicNetworkModel.

## Ethics statement

This study was waived by the Human Subject Research Ethics Review Committee at University of Tsukuba. Consent was waived for this study.

## Results

We evaluated the reproducibility of the Ubaldi et al. model, the Suda et al. model and the proposed model on empirical data using the previously described ten metrics. To generate data, we ran 20,000 iterations of each model, and used the first 20,000 pairs (caller-callee pairs) of empirical data. For each model, we tested all possible combinations of the parameters $1 \leq v$, $\rho \leq 30$ with 2 step intervals. The choice of strategy $s$ was selected based on the results of Ubaldi et al. [22]. For the APS Co-authors Network $s$ was set to ASW and for the Twitter Mentions Network $s$ was set to WSW. For the Suda et al. model, $s_c$ was set to PGBK [12]. For the

**Table 2. The parameter settings for the experiment. A blank space indicates the parameter is non-applicable to the model.**

| Model | $\rho$, $\nu$ | $s$ | $s_c$ | $\zeta$, $\eta$ |
|---|---|---|---|---|
| Ubaldi et al.(APS) | 2, 4, . . ., 30 | ASW | | |
| Ubaldi et al. (TMN) | 2, 4, . . ., 30 | WSW | | |
| Suda et al. (APS) | 2, 4, . . ., 30 | ASW | PGBK | |
| Suda et al. (TMN) | 2, 4, . . ., 30 | WSW | PGBK | |
| Proposed (APS) | 2, 4, . . ., 30 | ASW | | 0.0, 0.2, . . ., 1.0 |
| Proposed (TMN) | 2, 4, . . ., 30 | WSW | | 0.0, 0.2, . . ., 1.0 |

proposed model, all possible combination of $0 \leq \eta, \zeta \leq 1$ with 0.2 step intervals were tested. These parameter settings are summarized in Table 2.

Tables 3–5 show the best-fit parameter values of the Ubaldi et al. model, the Suda et al. model and the proposed model. Fig 2 illustrates the value of the distance $d$ between the empirical data and the models using the best fitting parameters. In both the APS Co-authors Network and the Twitter Mentions Network, the proposed model was found to capture the behavior of the empirical data more accurately than both existing models.

The results of the ten evaluation indicators in the existing and proposed models are depicted in Fig 3. In the proposed model, the novelty indicators $G$, $\langle h \rangle$, $R$, and $Y$ closely approximate the values of the empirical data in both the APS Co-authors Network and the Twitter Mentions Network. In particular, the recentness metric had the greatest improvement, showing that in the proposed model, nodes which enter the network later can receive significant attention. Although the youth coefficient $Y$ of the proposed model is slightly worse than that of the Ubaldi et al. model for the Twitter Mentions Network, the proposed model exhibited approximately the same or better accuracy than the existing models in the other indicators related to network topology.

The parameters $\rho$ and $\nu$ modulate the propensity to exploit the past and the propensity to explore the new, respectively. The ratio of $\rho/\nu$ can be used to gauge this propensity; when the value is large, the propensity to exploit a past callee is high, and when it is small, the propensity to explore a new callee is high.

Interpreting the fitted parameters from the proposed model, the APS Co-authors Network has a greater value of $\nu$ than $\rho$ ($\rho = 10$, $\nu = 16$, $\rho/\nu = 0.625$), indicating that each author actively accesses their adjacent possible space and co-authors papers with new authors at a higher rate than revisiting past authors. This trend is consistent with existing research by Ubaldi et al.

On the other hand, in the Twitter Mentions Network, $\nu$ has a value slightly closer to $\rho$ ($\rho = 18$ and $\nu = 24$, $\rho/\nu = 0.75$), indicating that new users are brought into the adjacent possible space at a similar rate as already mentioned users. This trend is also in line with the results of existing studies by Ubaldi et al.

Similarly, the parameters $\eta$ and $\zeta$ modulate the propensity to exploit the past and the propensity to explore the new, respectively, for the caller agents. This propensity can be gauged by

**Table 3. The optimal parameters for the Ubaldi et al. model.**

| | $\rho$ | $\nu$ |
|---|---|---|
| APS Co-authors Network | 1 | 25 |
| Twitter Mentions Network | 18 | 30 |

**Table 4. The optimal parameters for the Suda et al. model.**

|  | $\rho$ | $\nu$ |
|---|---|---|
| APS Co-authors Network | 2 | 20 |
| Twitter Mentions Network | 2 | 7 |

**Table 5. The optimal parameters for the proposed model.**

|  | $\rho$ | $\nu$ | $\zeta$ | $\eta$ |
|---|---|---|---|---|
| APS Co-authors Network | 10 | 16 | 0.2 | 0.6 |
| Twitter Mentions Network | 18 | 24 | 1.0 | 0.2 |

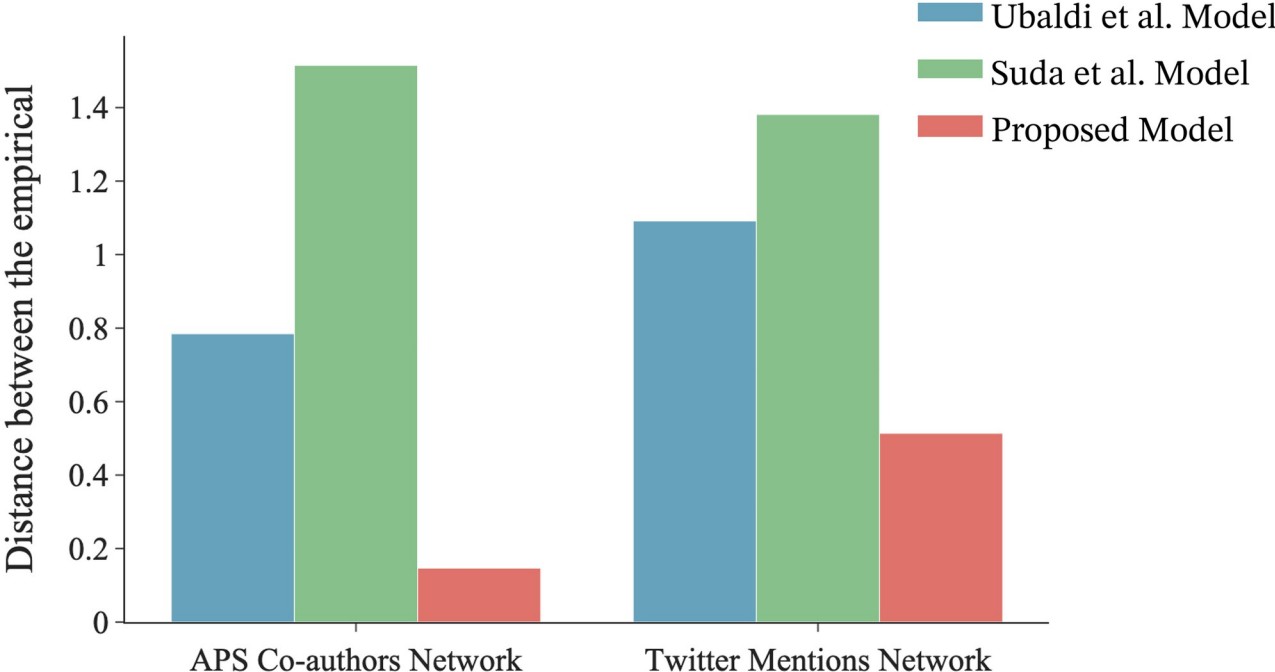

**Fig 2. Comparison of the distance between the existing models and the proposed model with the empirical data.** The distance $d$ is defined by expression 8, with smaller values indicating a better fit to the empirical data. Our proposed model outperforms both existing models.

the ratio $\zeta/\eta$; when the ratio is high, then the propensity to exploit a past caller is high and when it is low, then the propensity to explore a new caller is high.

For the APS Co-authors Network, $\zeta/\eta \approx 0.3$ ($\zeta = 0.2$ and $\eta = 0.6$), whereas $\zeta/\eta = 5$ ($\zeta = 1.0$ and $\eta = 0.2$) for the Twitter Mentions Network. These results suggest that the propensity to explore newer callers is much higher in the APS Co-author Network than in the Twitter Mentions Network. Overall, the results of the parameter estimates provide insights into the behavior of the network and how the proposed model is able to capture the attention dynamics of the network.

## Reproduction of a wave of novelty

The results demonstrated that our proposed model was able to reproduce the waves of novelty present in human activities with greater accuracy than the models proposed by Ubaldi et al.

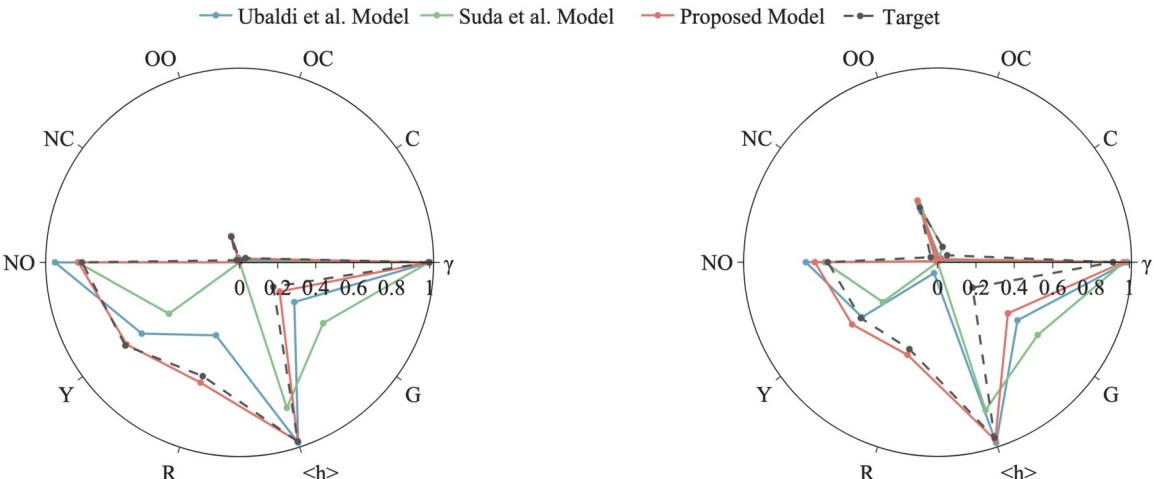

**Fig 3. Comparison of each measurement indicator between the existing and proposed models.** In both the APS Co-authors Network (a) and Twitter Mentions Network (b), the existing models reproduce the empirical data well for *NO, NC, OO, OC, C* and *γ*, which are indicators of the network structure. However, the existing models fail to capture the indicators for novelty waves, *Y, R* and *G*, which deviate significantly from the empirical data, with the exception of *Y* for the Twitter Mentions Network. The proposed model, on the other hand, greatly improves the indicators for novelty waves without significantly worsening the indicators on the network structure.

and Suda et al. We can quantitatively assess the waves of novelty in both the empirical data and the proposed model. For instance, as seen in Fig 4, the novelty waves in the APS Co-authors Network and the Twitter Mentions Network are depicted. The event time series of the data were divided into intervals of 200 steps along the x-axis, with the y-axis depicting the time when the most active agent in each interval was born. Fig 4(a) and 4(e) present the results for the empirical data, with a concentration of markers on the $y = x$ line indicating that the agents born in that interval are the most popular for their interval and that new agents are attracting more attention than old agents. In contrast, for both the Ubaldi et al. model Fig 4(b) and 4(f) and the Suda et al. model Fig 4(c) and 4(g), the markers are concentrated around the x-axis, indicating that the agents born in the early stages receive the most attention in each section, and the novelty wave is not reproduced. Our proposed model, as shown in Fig 4(d) and 4(h) reproduces behavior that more closely resembles the empirical data, with markers appearing on the $y = x$ line. This behavior leads to a significant improvement in the novelty indicators.

The behavior of individual agents can be further examined in more detail. Fig 5 illustrates the birth step and occurrence frequency of all the agents. If new agents are observed to attract more attention than old agents, as seen in the empirical data shown in Fig 5(a) and 5(e), the distribution exhibits significant variance. Conversely, the resulting distribution from the model proposed by Ubaldi et al. in Fig 5(b) and 5(f) and Suda et al. in Fig 5(c) and 5(g) is skewed in the $y = -x$ direction, where new agents are not able to gain attention over old ones. Although this skewness also occurs in the proposed model in Fig 5(d) and 5(h), for agents born after iteration 1000 the shape of the distribution is closer to the empirical data, for both the Twitter Mentions Network and the APS Co-authors Network. This indicates that the proposed model can more effectively reproduce the behavior of new agents gaining attention over old ones. Unlike the empirical data however, the most active agents in all three models are those that appeared at the start of the network history. This bias is a limitation of the model, in that agents appearing at the beginning of the simulation are highly likely to receive a lot of

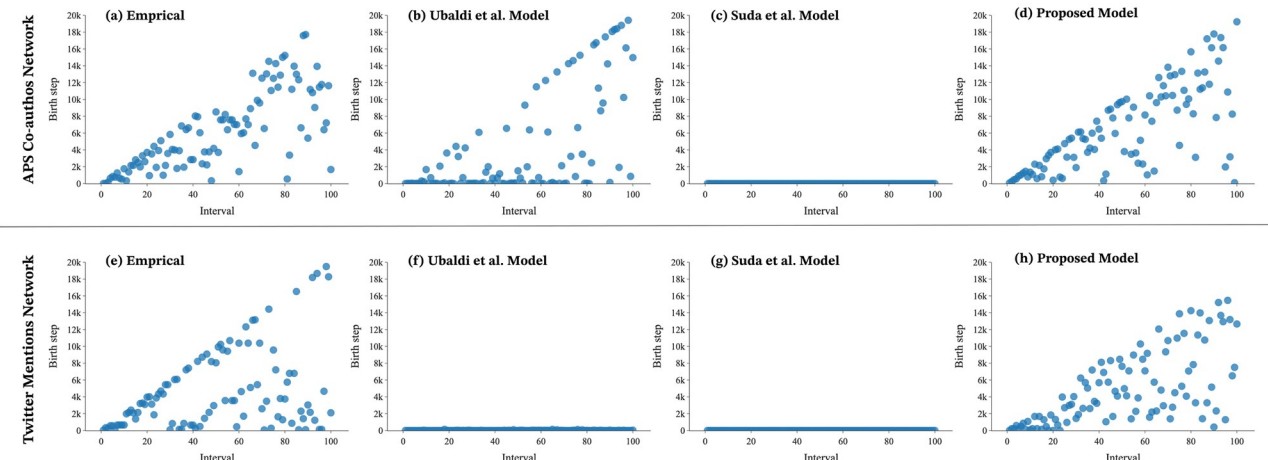

**Fig 4. The relationship between the agent that received the most attention during the interval and the birth step of that agent.** We define an interval by separating the iterations at regular intervals. Here 200iterations = 1interval. (a)(e): In the empirical data, the markers are distributed throughout the triangle, indicating that the agent's birth step differs depending on the interval. (b)(f): In the Ubaldi et al. model, markers are distributed unevenly around the $x$ axis. This indicates that, for many intervals, the birth step of the agent that received the most attention is biased towards early stages, suggesting that significant preferential selectivity is at play. (c)(g): In the Suda et al. model, markers are distributed exclusively on the $x$ axis. This indicates that the most active agent in every interval was introduced at the beginning of the network history. (d)(h): The proposed model mitigates the bias seen in the existing models and produces a distribution close to the empirical data.

attention. Despite this limitation, these results show that the proposed model can reproduce the empirical data with a higher degree of accuracy by generating waves of novelty, allowing newly created agents to gain significant attention.

The parameters $\zeta$ and $\eta$ play a crucial role in determining the behavior of the model through waves of novelty. The ratio $\zeta/\eta$ is used to control the emergence of waves of novelty.

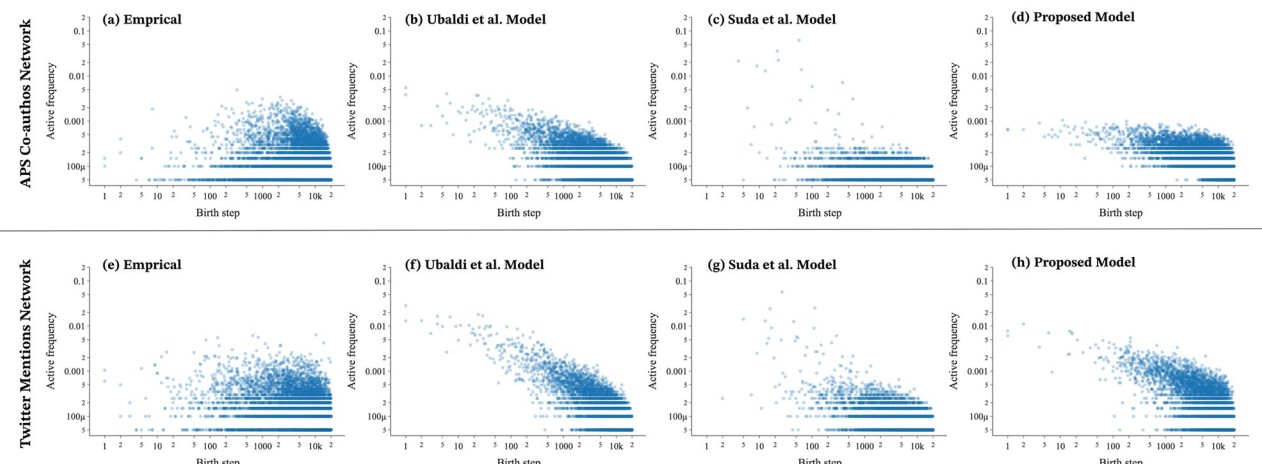

**Fig 5. Relationship between the iteration in which an agent was born and its active frequency.** (a)(e): In the empirical data, many markers are distributed in the top-right corner of the scatter diagram, indicating that agents born later can acquire high active frequency. (b)(f): In the Ubaldi et al. model, few markers are distributed in the top-right corner of the scatter plot, and agents born later are not able to gain attention. (c)(g): Similarly, in the Suda et al. model, few markers are generated near the top-right of the scatter plot. (d)(h): The proposed model has more markers distributed on the right side of the scatter plots, indicating that some of the agents born later attract significant attention, as in the empirical data. On the other hand, all three of the models have points on the top-left of the graph and not on the bottom-left of the graph, as in the case with the empirical data. This means that early agents in the models always gain a large amount of attention, whereas in reality, agents who appear early in the social network may not receive a lot of attention. This is a limitation of these models.

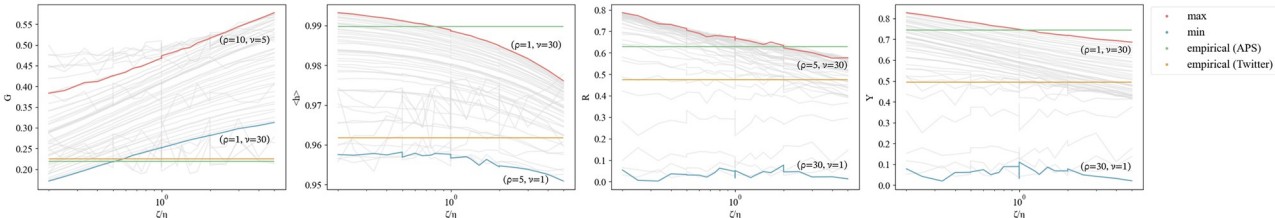

**Fig 6. The relationship between the parameters $\zeta/\eta$ of the proposed model and the indicators $G$, $\langle h \rangle$, $R$, and $Y$ for the novelty waves.** The gray lines in the plot represent the range of values for $\eta$ and $\zeta$ ranging from 0.1 to 1.0, with a 0.2 step size. The paired values of $\rho$ and $v$ range from 1 to 30 with a step size of 5, which results in a total of 49 gray lines. The red lines indicate the pair $(\rho, v)$ which takes the absolute maximum value for some $\zeta/\eta$. Similarly, the blue lines indicate the pair $(\rho, v)$ which takes the absolute minimum for some pair $(\zeta/\eta)$. Each line is the average of 10 runs. The proposed model is able to adapt to a variety of domains by adjusting its parameters. The ratio of $\zeta$ to $\eta$ determines the balance between selecting newer or older agents as the callee. A smaller ratio leads to a higher proportion of newer agents being chosen, while a larger ratio favors the selection of older agents through preferential attachment. As the ratio increases, the number of occurrences for $G$ tends to rise, leading to a greater disparity in the distribution. Conversely, the values of $Y$ and $R$, which tend to be higher when there is a higher frequency of new nodes, decrease as the ratio increases. Additionally, the diversity of interacting nodes, as measured by $\langle h \rangle$, tends to be lower when the ratio is higher and newer agents are more frequently selected.

Fig 6 illustrates the relationship between $\zeta/\eta$ and the four metrics for assessing novelty ($G$, $R$, $\langle h \rangle$, and $Y$). The gray lines in the plot represent the range of values for $\eta$ and $\zeta$ ranging from 0.1 to 1.0, with a 0.2 step size. The paired values of $\rho$ and $v$ range from 1 to 30, with a step size of 5, which results in a total of 49 gray lines. The red lines indicate the pair $(\rho, v)$ which takes the absolute maximum value for some $\zeta/\eta$. Similarly, the blue lines indicate the pair $(\rho, v)$ which takes the absolute minimum for some pair $(\zeta/\eta)$. Each line is the average of 10 runs. The proposed model is able to adapt to a variety of domains by adjusting its parameters. The yellow and green lines show the values found in the two empirical datasets.

As the ratio increases, the number of occurrences for $G$ tends to rise, leading to a greater disparity in the distribution. Conversely, the values of $Y$ and $R$, which tend to be higher when there is a higher frequency of new nodes, decrease as the ratio increases. Additionally, the diversity of interacting nodes, as measured by $\langle h \rangle$, tends to be lower when the ratio is higher and newer agents are more frequently selected. These trends are consistent with the previous work by Monechi et al. [19]. The figure illustrates the importance of balancing the strength of preferential attachment and the search for novelty in the proposed model, which is crucial for reproducing the behavior of the empirical data.

## Discussion

### Principle for generating waves of novelty

The proposed model classifies agents into five classes based on their labels and appearance. The model assigns different weights to each class, thereby influencing the selection probability of callers, which previously relied solely on preferential attachment in the existing agent-based model proposed by Ubaldi et al. The introduction of classes leads to the generation of waves of novelty. By analyzing the changes in the number of agents and selection probabilities for each class, it is possible to understand how they contribute to novel agents gaining attention beyond preferential attachment. To illustrate this, the proposed model was run for the optimal parameters as shown in Table 5, for both the APS Co-authors Network and the Twitter Mentions Network. Figs 7 and 8 show how the number of agents and the selection probability of each class changes throughout the course of the simulation.

Class 1 represents agents with urn size 0, that is, agents that exist only in the adjacent possible space and have not been selected. The number of agents in class 1 increases with each

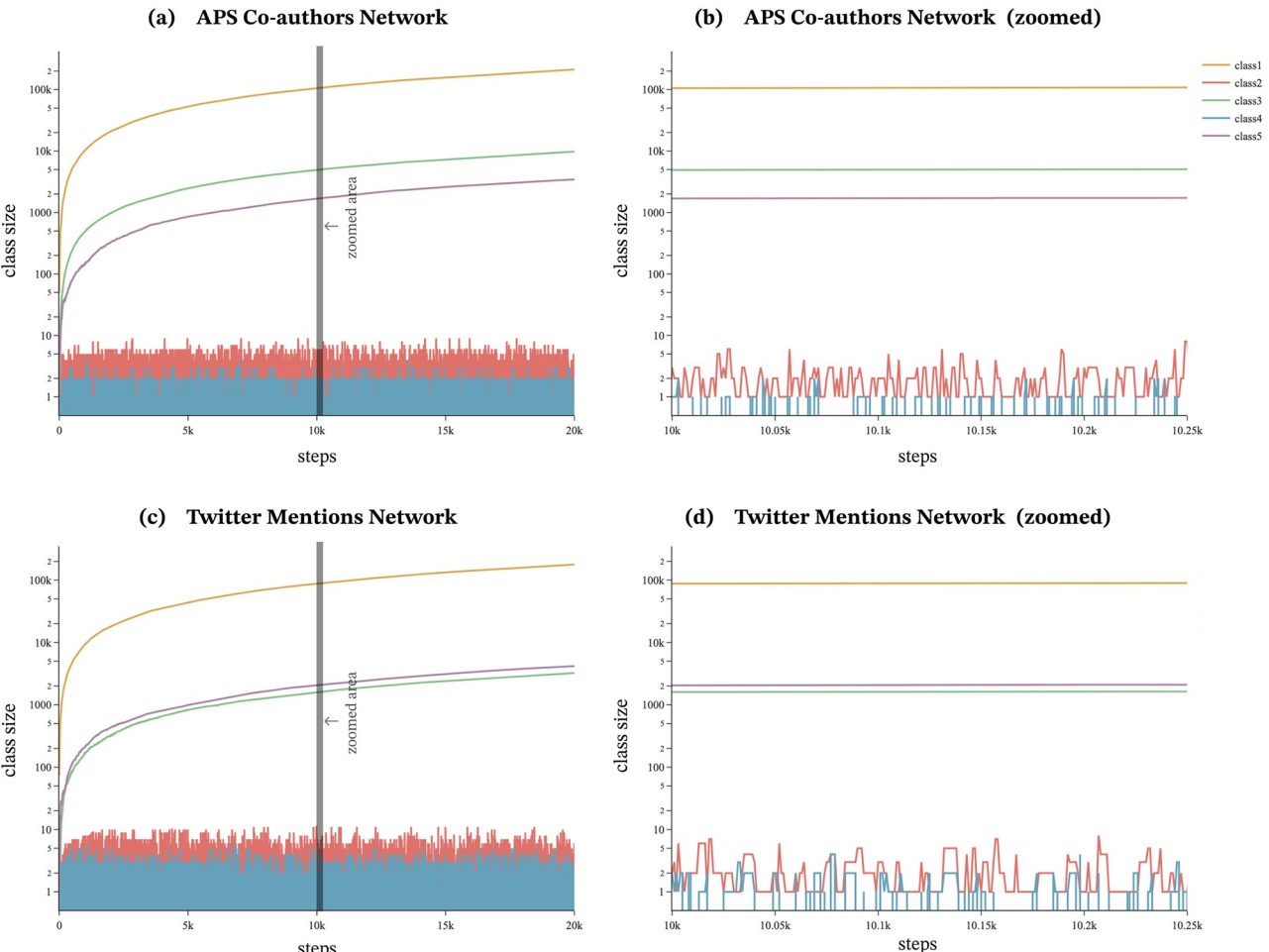

**Fig 7. The number of agents belonging to each class.** (a)(c): Classes 2 and 4 are agents with the label $L_{t-1}$, and each of these classes have at most $v + 1$ number of agents. The fluctuations of these class sizes are further highlighted in the enlarged figures (b) and (d). On the other hand, the number of agents in class 0, 1, and 3 continues to increase.

iteration, as illustrated by the yellow line in Fig 7. However, their selection probability is always 0, and as a result, they are not selected as caller agents (not plotted in Fig 8).

Class 2 represents a set of agents that have been previously selected as caller agents and possess the same labels as the agents selected in the previous iteration. In other words, class 2 refers to agents which share the same "parent". As the number of agents in this class is capped at $v + 1$, the number of individuals within class 2 is relatively small. Fig 7 illustrates that the number of class 2 agents (depicted by the red line) fluctuates without exceeding $v + 1$ in both the APS Co-authors Network and the Twitter Mentions Network. However, the agents belonging to this class are given the highest weight, 1, which gives them a higher selection probability than agents in other classes, as shown in Fig 8.

Class 3 comprises of agents that have been previously selected as caller agents and possess a label different from the one selected in the previous iteration. The population of agents belonging to class 3 increases over time, as demonstrated in Fig 7 in green. The selection probability of agents from class 3 fluctuates around a constant value, as depicted in Fig 8, because the selection probability of class 3 is given by $\zeta f$, and this value decreases as the number of $N_{\overline{L_{t-1}}}$

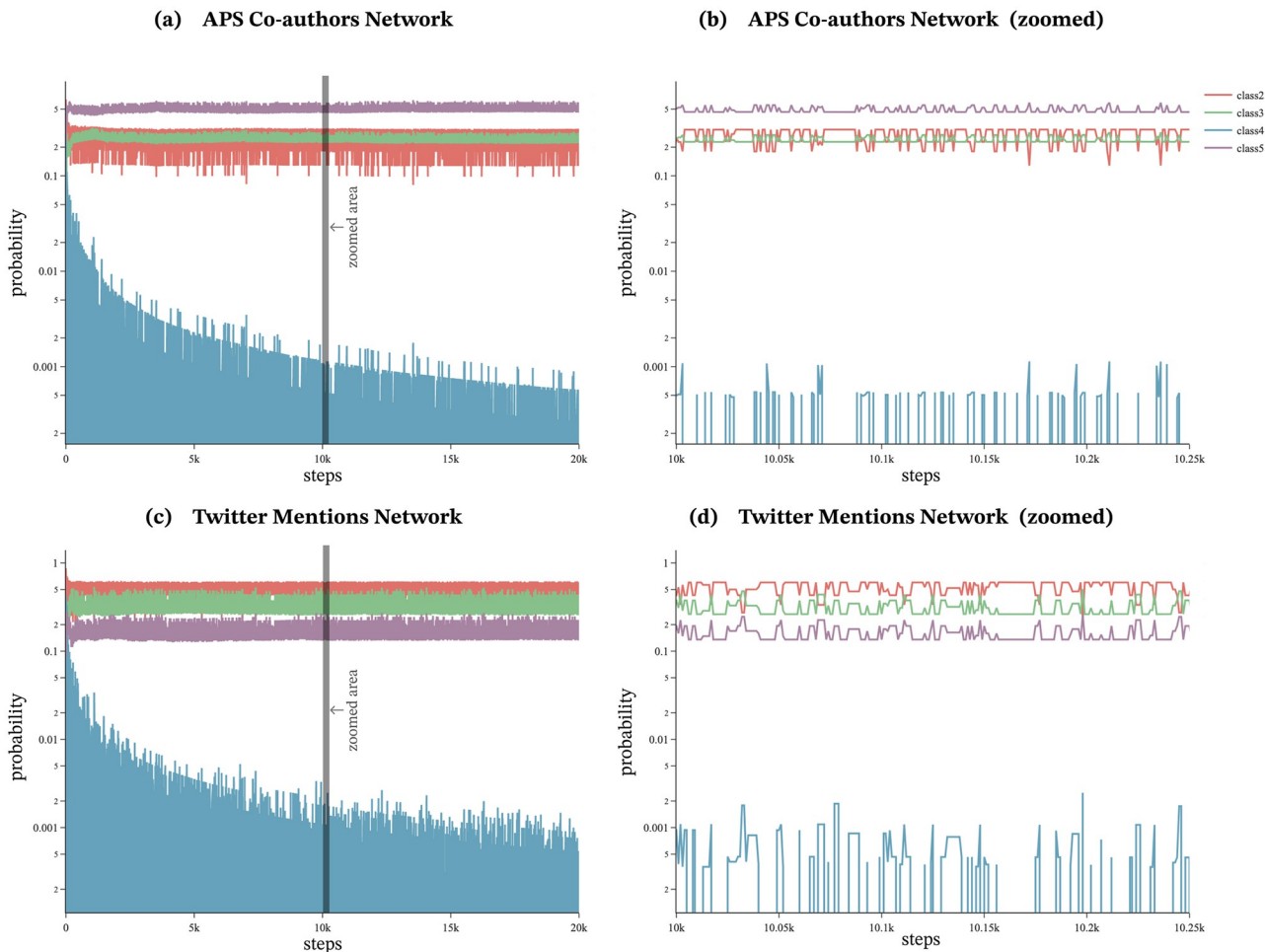

**Fig 8. Class selection probabilities.** (a)(c): Classes 2, 3, and 5 maintain stable selection probabilities with fluctuations. (a) Class 5 has the highest selection probability in the APS Co-authors Network, whereas (c) class 5 has the lowest selection probability in the Twitter Mentions Network, suggesting that an attention difference exists for novel agents in the two datasets. (b)(d): The enlarged figures (b) and (d) show that the selection probabilities of class 2 and class 3 are interchangeable. This alternation of probabilities leads to *waves of novelty*.

increases, thus offsetting the increase in the number of agents in class 3. The selection probability of agents from class 2 and class 3 oscillates alternately at close values, as shown in Fig 8.

Classes 2 and 3 consist of agents with labels that have been previously selected as caller agents, whereas those belonging to classes 4 and 5 have never been selected as caller agents. Among these, agents belonging to class 4 possess the same label as the one selected in the previous iteration, while those belonging to class 5 possess different labels. The number of individuals in class 4 is capped at $v + 1$, similar to class 2, and thus the number of agents in class 4 oscillates up to $v + 1$, as depicted in Fig 7 in blue. The selection probability of agents in class 4 gradually decreases as shown in Fig 8.

Agents belonging to class 5 have never been selected as caller agents and possess labels different from the one selected in the previous iteration. The number of agents in class 5 increases over time, as depicted in Fig 7 in purple. The selection probability of class 5 fluctuates around a specific value, as shown in Fig 8. The higher probability of class 5 indicates a higher likelihood of newer agents being selected. Additionally, the larger the value of $\eta$, the higher the

selection probability of class 5. Indeed, the APS Co-authors Network, with higher values of $\eta$ ($\eta = 0.6$), has a higher selection probability than the Twitter Mentions Network ($\eta = 0.2$). The consistently high selection probability of class 5 indicates that the selection probability of novel agents, who have never been selected as callee agents, has increased which contributes to the generation of novelty waves by the proposed model.

Fig 9(a) illustrates the transition of a single agent (depicted in red in Fig 9(b)) between classes 1, 2, 3, 4, and 5, and its cumulative number of selections, visualizing the process through which the agents attract attention beyond preferential attachment in the Twitter Mentions Network. The colors of the markers correspond to classes 1 to 5 (class 1: yellow, class 2: red, class 3: green, class 4: blue, class 5: purple). The marker ▲ denotes selection as a caller, the marker ▼ denotes selection as a callee and the marker • denotes inactivity. We observe that the agent is selected sequentially as a caller when it belongs to class 2. This behavior is captured as the most selected agent in the corresponding interval, as depicted in Fig 9(b). We also observe this agent being selected as a callee sequentially when the agent belongs to class 3. This is caused by interactions among agents and the agents not being bursty enough to be captured as the most selected agent in the corresponding interval. This result confirms that changes in the selection probability of callers play an essential role in the generation of waves of novelty.

## Limitations and future work

The proposed model effectively reproduces waves of novelty by extending the agent-based model proposed by Ubaldi et al. However, there are certain limitations that remain unaddressed. For instance, as shown in Fig 5, in empirical data, some agents that were created relatively early have a low attention span (bottom-left region of the scatter diagram). While the proposed model reproduces the behavior of newer agents gaining attention beyond preferential attachment, it fails to reproduce the behavior of agents that were born early but only gained low attention. The value of the Gini-like coefficient ($G$), which measures the disparity in agents' attention, is also higher for the values fitted by the proposed model than for the empirical data.

The proposed model assumes that many early-born agents only generate behavior that obtains great attention, resulting in a higher value of $G$ than what is observed in the empirical data. Some early-born agents in the empirical data fail to gain attention, but this behavior is not captured by the proposed model. If this behavior is considered, it would allow the proposed model to disperse agents' attention more evenly and bring the value of $G$ closer to that of the empirical data.

One potential avenue for further improving the proposed model and better reproducing the behavior of empirical data is by modifying the selection mechanism of caller agents across classes. In the current proposed model, all agents belonging to the same class are assigned the same weight, and the selection of agents is determined randomly. However, other selection methodologies other than random selection can be utilized. For example, the last agent added could always be selected instead of a randomly selected agent from a class, which would increase the probability of selecting newer agents. Alternatively, increasing the selection probability of agents that interact with more influential agents, could have a higher chance of being selected as well. This would allow agents that have not previously attracted attention to attract attention by leveraging the influence of powerful agents. Indeed, previous studies have reported that adding such biases when selecting a caller agent reproduces behavior closer to the empirical data [12].

In addition, it has been shown by Ubaldi et al. that the choice of memory buffer exchange strategy $s$ greatly affects the resulting network structure [22]. In this work, we only examined

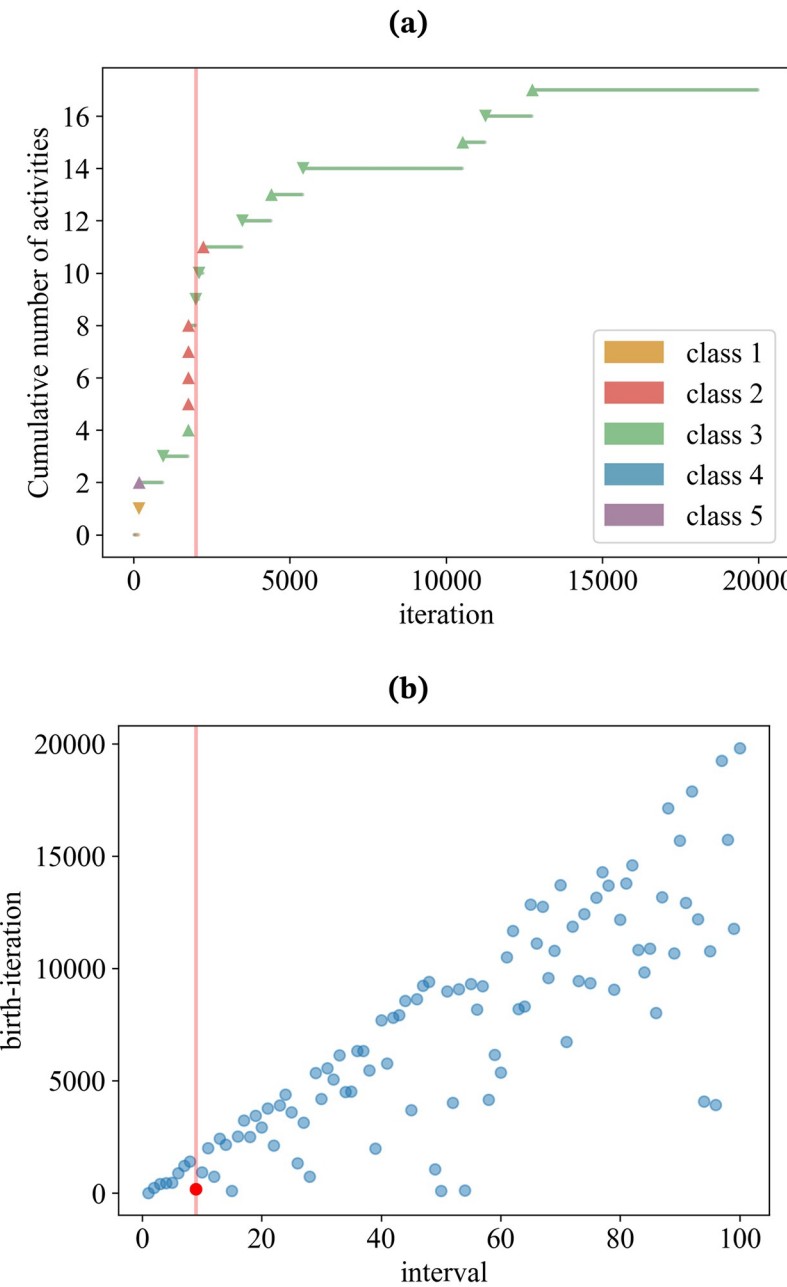

**Fig 9. The relationship between the cumulative number of activities and the class of affiliation for a given agent extracted from the model.** (a) illustrates the transition of a single agent throughout the simulation. The marker ▲ denotes selection as a caller, the marker ▼ denotes selection as a callee and the marker • denotes inactivity. At the beginning of the simulation this agent does not yet exist in the adjacent possible space of any active agent and does not yet have an urn. At approximately iteration 10 the agents urn is created when it enters the adjacent possible space of an active agent. Shortly after this, the agent is selected as a callee for the first time, thus being promoted to class 5. Soon after, the agent is selected as a caller for the first time and enters class 3. At approximately iteration 2000, the agent is selected as a caller again, but this time have the label $L_{t-1}$, thus being promoted to class 2. At this point the agent continues to gain attention, being selected as a caller numerous times. This sudden burst in attention can be attributed to the the high probability of caller selection associated with being in class 2. This spike in popularity is made apparent in (b), with the red dot showing the agent we are analysing in (a). At approximately iteration 2000, this agent is the most active agent in the time interval, indicating that it has triggered a wave of novelty. After this burst in activity, the agent enters class 3. Although entering class 2 once more at iteration 2500, for the remainder of the simulation the agent remains in class 3.

the optimal strategies found by Ubaldi et al., however, it would be interesting to view how the choice of strategy *s* affects the waves of novelty. In future work, one could examine how the network metrics are affected as *s* is changed to other strategies, including the PGBK strategy introduced in [12], or the vectorisation of strategies as introduced in [27].

Finally, the models discussed throughout this work have concerns regarding mind reading. To highlight this issue, consider the situation where there are three agents: A, B and C. In all three of these models, A is able to recommend B to C even if A and B have never interacted. That is, the memory buffer of A can contain B, even if B only exists in the adjacent possible space of A. One possible solution to this is to limit the memory buffer to only contain the agents in the actual space. This may lead to more accurate modelling of real data and is left to future work.

## Acknowledgments

The authors would like to thank Naoki Maejima for the insightful discussion of our model and data analysis.

## Author Contributions

**Conceptualization:** Mikihiro Suda, Takumi Saito, Mizuki Oka.

**Data curation:** Mikihiro Suda, Nanami Iwahashi.

**Formal analysis:** Mikihiro Suda, Ciaran Regan.

**Funding acquisition:** Mizuki Oka.

**Investigation:** Mikihiro Suda, Takumi Saito, Nanami Iwahashi, Ciaran Regan, Mizuki Oka.

**Methodology:** Mikihiro Suda, Takumi Saito, Mizuki Oka.

**Project administration:** Mizuki Oka.

**Resources:** Mikihiro Suda, Mizuki Oka.

**Software:** Mikihiro Suda, Nanami Iwahashi, Ciaran Regan.

**Supervision:** Mizuki Oka.

**Validation:** Mikihiro Suda, Takumi Saito, Ciaran Regan, Mizuki Oka.

**Visualization:** Mikihiro Suda, Nanami Iwahashi.

**Writing – original draft:** Mikihiro Suda, Mizuki Oka.

**Writing – review & editing:** Mikihiro Suda, Ciaran Regan, Mizuki Oka.

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
