## [Decision Letter · Decision Letter 0]

14 Apr 2023

PONE-D-23-01074Simulating Emergence of Novelties Using Agent-Based ModelsPLOS ONE

Dear Dr. Oka,

Thank you for submitting your manuscript to PLOS ONE. After careful consideration, we feel that it has merit but does not fully meet PLOS ONE’s publication criteria as it currently stands. Therefore, we invite you to submit a revised version of the manuscript that addresses the points raised during the review process.

Please, take into account all the reviewers' comments while preparing the revised version of the manuscript. Moreover, be sure that data underlying your findings are fully available for the sake of reproduction.

We look forward to receiving your revised manuscript.

Kind regards,

Letterio Galletta

Academic Editor

PLOS ONE

Journal Requirements:

"This work was supported by JSPS KAKENHI Grant Numbers 19H04214,20H04163, 21H03414."

"NO authors have competing interests."

Reviewers' comments:

Reviewer's Responses to Questions

**Comments to the Author**

1. Is the manuscript technically sound, and do the data support the conclusions?

Reviewer #1: Yes

Reviewer #2: Yes

2. Has the statistical analysis been performed appropriately and rigorously? 

Reviewer #1: Yes

Reviewer #2: Yes

3. Have the authors made all data underlying the findings in their manuscript fully available?

Reviewer #1: Yes

Reviewer #2: No

4. Is the manuscript presented in an intelligible fashion and written in standard English?

Reviewer #1: Yes

Reviewer #2: Yes

5. Review Comments to the Author

Reviewer #1: The proposed work builds upon existing Polya's urn innovation models to come up with a new model that better reproduces the occurrences of wave of novelties.

I would ask the Authors to address the following point prior to publication.

In a Polya's urn model the content of the urn is hidden.

What we measure is the stream observed as the result of picking up elements from the urn.

All the quantities related to innovation, Heaps', Zipf's and Taylor's laws are determined from the stream not from the urn.

If there is only one urn I do not see any conceptual problem.

When we have different urns associated to different agents, we should be careful to avoid mind reading.

We are not allowed to look into other's head, the urn in this case.

In iteration one, for example, agent A adds \\nu+1 new agents that are in the urn of B and that were never observed.

Agents E and F are in B's head and not accessible.

They are in the adjacent possible of B, so that even B has never observed them.

How do you avoid mind reading? Can you modify the model to solve this problem?

If not, the problem has to be mentioned and well explained in the limitations section.

Best regards

Reviewer #2: The authors of this work present a very interesting extension of an existing model by Ubaldi et al to mimic how a social network grows. While the previous model foresees preferential attachment, with the initial agents being the most influential and active, in the proposed model new agents can be more active than the first individuals over a short period of time, leading to waves of novelties. The authors successfully show that the proposed model better resembles the data from APS Co-authors Network and Twitter Mentions Network with respect to the model by Ubaldi et al. on a number of topological and dynamical metrics.

Although the paper is well structured and the results of the proposed model are absolutely worth publishing, in my opinion there are a few points that the authors should address before it can be published. Apart from minor problems here and there, the authors should better define their work with respect to existing literature, and better explain the mechanisms behind the new model. Although the dynamics is solid and works, interpretations of the chosen dynamics are missing. This is fundamental to make this work be understood, further improved or applied to other contexts by the broader scientific community. As a matter of fact, given the fact that the proposed model manages to reproduce the waves of novelties in the data as the authors claim, it is not clear what is the underlying mechanism that they introduce, which is not discussed enough. This in turn creates some problems in justifying what has been written at the end of the abstract about how this could have implications for the design of effective interventions on social networks.

Moreover, they do not make comparisons of their model with another model generalizing the model by Ubaldi et al in the same direction ("Agent-based model using adjacent possible space for generating waves of novelty"), which already improves the results towards the same goal, using a similar analysis. Here, this other paper is shown only as a reference to justify the fact that a different mechanism with a type of bias should be included in the model. Notice that this other model has been proposed by the same authors. Finally the authors do not provide code or data, so it is not possible to check if the results are solid.

More detailed comments below.

MAJOR ISSUES

-1- Lines 55-58. Here you mention your previous work [20], in which you already change the rule to choose the caller. In that previous paper you have already found a model which has “created asymmetries in the semantics, thereby leading to waves of novelties”. The results you obtained there leveraged on the same type of analysis. In this new paper, you are again proposing a modification of the caller mechanism to reproduce waves of novelties, this time introducing labels. In order for this paper to be original and substantial new work, you need to better describe what you have done in the other paper, and what maybe has failed there and you are instead recovering in this paper. Otherwise it seems just an alternative solution to the same problem that has already been solved in another paper, and as such it should be stated somewhere. With the same argument, I would like to see a comparison between these two modifications of the model, so that the reader understands what is the improvement brought by the model proposed here with respect to not only the original model by Ubaldi et al, but also the generalized ABM proposed in the cited paper. This is even more important considering how, in the discussion, you mention that including biased strategies like the one in the cited paper can improve the results. If the authors decide not to compare these models, which I recommend against, they should sufficiently explain why and provideat least a sufficient analysis of the difference between the two models.

-2- Proposed model description and figure1 (Lines 156-191). There are some things that are not clear to me.

- First of all, in your model, it seems like N_L is always nu+1, since this is the number of new agents triggered by calling an agent for the first time and to which you assign the same label. Is so, it should be stated.

- Then, by looking at the example of the figure, when you count the number of agents not with label L, you are considering only agents that have been called at least once before. For example, in iteration 3, you are not counting C, D, F, G, H. So basically this counts the total number of activated agents in the system, minus the nu+1 agents sharing the same label L. It this correct? You might want to clarify all these points in the text.

- Moreover, it is not clear to me why you choose such functions f and g, and why you choose those weights in the various classes. Can you add an intuitive explanation of these functions? For example, for nodes in class 3 you assign weight eta*f, i.e., \\frac{\\eta N_L}{N_L + \\eta (N-N_L)}. Can you explain the role of \\eta and of f in this weight function? Point is that these new weights might work perfectly looking at the results, but if they have no connection to the phenomena you want to describe, then the model has a low value.

- What is the "physical" mechanism controlling the weights you have introduced? What happens, for example, when eta is 0 or 1? Same for zeta and the other weights. More in general, you should a deeper explanation of why you divide the possible new callers into these classes that depend on the label of the last caller.

- In line 183 you talk about the two parameters effect on the propensity to exploit vs explore, but it is not clear how they do so, based on the model definition. This is later explained in the results, but it is important to provide a first interpretation of the possible outcomes based on the model definition. Can you state if you tend to explore more with higher or lower values of each parameter, giving some examples of how the weights would be modified?

MINOR ISSUES

Line 12-15 Original Polya’s urn with two colors (black and white) does not reproduce these laws, which foresee a power-law behavior. This is instead obtained when considering a very large number of initial colors or when considering generalization of the Polya's urn, like the ones you later cite. In my opinion, citations 14-18 are not so appropriate to what you have stated in this paragraph, unless you meant to say something else that I have misinterpeted. For example, the Matthew effect you cite does not talk about Zipf or Heaps, nor Polya’s urn.

Line 36 “Kaufman” typo, missing f

Line 38-39 Paragraph is not clear, and I don’t understand what “respectively” refers to here.

Line 40-43 You are repeating what said in lines 38-39, but using less precise terminology. It is not clear what an agent is in the context of the system/urn, since you had not talked about an agent before. In fact, the growing system can be anything, could be for example the knowledge of a person or thenumber of inventions of a population, with balls representing keywords, songs, patents, etc. You need to clearly state here that in this work you are going to study a social system, and that for you the urn represents the space of possible contacts of an individual, and each color is a person. As a matter of fact, you make the assumption ball=agent throughout the paper. This specific interpretation is key in Ubaldi et al, but not in the original paper by Tria et al. You might want to say first that Ubaldi et al. extended the urn model with triggering to reproduce the growth of a social system, considering different urns, representing the space of possibilities of different agents, with the color of balls representing various agents. And then you can further notice that the ball itself does not represent the agent. Rather, the color or ID of the ball represents the agent. In fact, there can be many balls with the same color/ID representing the same agent. Since all this is the base of the results, it should as clear as possible for the reader.

Line 120-124. The nu+1 new balls are added only if the pair (caller, callee) is activated for the first time. All subsequent times, they only reinforce with rho balls of the other. I hope you have not done this mistake also while coding, but I cannot check since no code has been provided. Please share the code for open reproducibility of your results.

Figure 1 typos: top left panel “an possible relation”. Repeatedly said parter instead of partner. Top right there is a smaller table class1 to weights table to be removed. Iteration 2 in the network E should be colored. Iteration 3 the numbers in f and g are wrong (0s should be 1)

Line 212 Cluster coefficient is more commonly referred to as clustering coefficient.

Line 259: maybe you wanted to say interval I_i?

Line 259-260: the way it is phrased, you are considering the node with the highest number of interactions in the whole growth time step. I suppose you instead meant the highest number of interactions in the interval I_i, like in Monechi et al.

Line 262. Forgotten to add the paragraph label Local entropy?

Line 266-267: maybe you wanted to say “node j”?

Line 282 What are node species?

Line 293. In this distance metric, you should say how each property is normalized. If the measures do not take values in the whole spectrum [0,1], then it should be clearly stated. For example, in the definition of OC, OO, NC and NO in line 228-229, you say that these are the frequencies, not necessarily normalized, so could be huge values.

Line 295-297: also include the other two parameters you are adding in the proposed model

Line 313-320 and Table 1. The parameters shown in Table 1, in particular nu, are outside the range you state in these lines. So which parameters are you using for the original model? If the maximum value considered is nu = 20, you should actually increase it further, because you are finding a best fit at the edge of the space of the parameters, showing that the best fit could be outside the set of parameters you have tested.

Line 318-320: As for the strategy s, why have you chosen the strategy SSW? You have stated elsewhere that the strategy chosen can influence quite a lot the way the network grows and I agree. As reported in the original paper by Ubaldi et al, they found the best strategy to be ASW for the APS subsample and WSW for the Twitter Mentions Network. Have you tried other strategies? Does the choice of the strategy change the results? If so, how? Mention this in the limitations of the study or provide further evidence.

Line 354 typo in zeta

Figure 4 and 5. They seem to be interchanged, since in the text you refer to Fig 4, but it is Fig5.tif

Lines 378-390 and related figure. This figure is very hard to read and could be misleading. For example, I could say that also for the proposed model there is some skewedness, where higher active frequencies are found only in low birth times, which are not seen in the empirical distribution. You should state this issue. Moreover, the panels have different scales and limits in the y axis. This does not help in the interpretation. You could add that in the existing model there are some points with very high frequency (higher than the highest in the empirical), and are all among the first birth steps. After 100 steps, the distribution seems very similar to the proposed model. Moreover, the scatter points are all the same, and it is hard to understand the density of points. You could use a color scale for the points, depending on the density of points, add an alpha, or consider some binning or cumulated distributions. This could help visualize better the data and interpret it.

Line 396-397 How do you define the maximum and minimum values among all the lines? For example in the related figure it seems there are other gray lines above the red line or below the blue one. Also, to which combination of nu and rho they refer to?

Figure 7 and 8. Better to delete the "(log scale)" from the ylabel, because otherwise one can interpret the numbers in the y axis to be considered in logarithm, i.e., you are plotting the logarithm of the class size. So for example in class 1 at 20k, it could be interpreted as log(class_size) = 100k.

Furthermore, for better clarity, you should do the zoom for the cluster size as you did in the other figure. This should help to see how class 2 + class 4 is always nu+1

Line 412-422 Here are you referring to a specific model simulation? With which parameters?

Line 427 missed closed parenthesis

Line 468-470 Add that a circle represents a time step in which the selected agent is not active.

Figure 9 caption. There are some mistakes and incomplete. The fact that it has a sudden increase does not indicate that it is classified as class 2, rather that as soon as it called, it is has a burst in activity as a caller (blue triangles up). After this it remains in class 3 and is only called some other few times. Also, you should mention and explain what you are plotting in panel b in the caption.

6. PLOS authors have the option to publish the peer review history of their article (what does this mean?). If published, this will include your full peer review and any attached files.

Reviewer #1: No

Reviewer #2: No

---

## [Author Response · Author response to Decision Letter 0]

27 Sep 2023

We have responded to all the review comments in the "Response to Reviewers.pdf". 

Please refer to the PDF.

---

## [Decision Letter · Decision Letter 1]

30 Oct 2023

Simulating Emergence of Novelties Using Agent-Based Models

PONE-D-23-01074R1

Dear Dr. Oka,

We’re pleased to inform you that your manuscript has been judged scientifically suitable for publication and will be formally accepted for publication once it meets all outstanding technical requirements.

Kind regards,

Letterio Galletta

Academic Editor

PLOS ONE

Additional Editor Comments (optional):

Reviewers' comments:

Reviewer's Responses to Questions

**Comments to the Author**

1. If the authors have adequately addressed your comments raised in a previous round of review and you feel that this manuscript is now acceptable for publication, you may indicate that here to bypass the “Comments to the Author” section, enter your conflict of interest statement in the “Confidential to Editor” section, and submit your "Accept" recommendation.

Reviewer #1: All comments have been addressed

Reviewer #2: All comments have been addressed

2. Is the manuscript technically sound, and do the data support the conclusions?

Reviewer #1: Yes

Reviewer #2: Yes

3. Has the statistical analysis been performed appropriately and rigorously? 

Reviewer #1: Yes

Reviewer #2: Yes

4. Have the authors made all data underlying the findings in their manuscript fully available?

Reviewer #1: Yes

Reviewer #2: Yes

5. Is the manuscript presented in an intelligible fashion and written in standard English?

Reviewer #1: Yes

Reviewer #2: Yes

6. Review Comments to the Author

Reviewer #1: My only concern was the "mind reading" aspect in the modeling scheme of this article as well as in previous literature.

The added statement in the parer in this regard might help to raise awareness in these kinds of problems that are usually overlooked in complex system studies applied to social sciences.

I recommend this paper for publication.

Best regards

Reviewer #2: Thanks to the authors for the nice reading. They have addressed all my previous comments, and the manuscript is ready to be published, prior to once again double check typos and gramatical errors that I have still found. For example (not an exhaustive list, please double check again)

Line 164 (in an attempt TO reduce)

191 (will all assigned -- gramatically wrong.. "will all be assigned to the same label")

206 (has BEEN selected)

227.. (This choice OF f.. IS motivated ...., who have shown that with this choice, the growth of the number of distinct elements with time t follows the Heaps' law)

322 "Similarly"

500 (was RUN)

598 (only exists IN the adjacent possible space)

7. PLOS authors have the option to publish the peer review history of their article (what does this mean?). If published, this will include your full peer review and any attached files.

Reviewer #1: No

Reviewer #2: **Yes: **Gabriele Di Bona

---

## [Editor Report · Acceptance letter]

30 Nov 2023

PONE-D-23-01074R1 

Simulating Emergence of Novelties Using Agent-Based Models 

Dear Dr. Oka:

I'm pleased to inform you that your manuscript has been deemed suitable for publication in PLOS ONE. Congratulations! Your manuscript is now with our production department. 

Kind regards, 

on behalf of

Dr. Letterio Galletta 

Academic Editor

PLOS ONE